Expanding standards in viromics: in silico evaluation of dsDNA viral genome identification, classification, and auxiliary metabolic gene curation

Pratama Akbar Adjie 1 2
Bolduc Benjamin 1 2
Zayed Ahmed A. 1 2
Zhong Zhi-Ping 1 2 6
Guo Jiarong 1 2
Vik Dean R. 1 2
Gazitúa Maria Consuelo 3
Wainaina James M. 1 2 7
Roux Simon sroux@lbl.gov 4
Sullivan Matthew B. sullivan.948@osu.edu 1 2 5
1 Department of Microbiology, Ohio State University , Columbus , OH , United States of America
2 Center of Microbiome Science, Ohio State University , Columbus , OH , United States of America
3 Viromica Consulting , Santiago , Chile
4 DOE Joint Genome Institute, Lawrence Berkeley National Laboratory , Berkeley , CA , United States of America
5 Environmental and Geodetic Engineering, Ohio State University, Department of Civil , Columbus , OH , United States of America
6 Byrd Polar and Climate Research Center, Ohio State University , Columbus , OH , United States of America
7 Infectious Diseases Institute at The Ohio State University, Ohio State University , Columbus , OH , United States of America
Lefkowitz Elliot
Electronic publication date: 2021 Jun 14
Publication date: 2021
Volume: 9
Electronic Location ID: e11447
Received 2020 Nov 27; Accepted 2021 Apr 22
Copyright: ©2021 Pratama et al.
Copyright year: 2021
Copyright holder: Pratama et al.
License: This is an open access article distributed under the terms of the Creative Commons Attribution License, which permits unrestricted use, distribution, reproduction and adaptation in any medium and for any purpose provided that it is properly attributed. For attribution, the original author(s), title, publication source (PeerJ) and either DOI or URL of the article must be cited.
License URL: https://creativecommons.org/licenses/by/4.0/

Keywords: Benchmarks, Standard operating procedure, Viruses, Viromics, Ecology

Funding: NSF OCE1829831 ABI1758974 US Department of Energy DE-SC0020173 248445 Gordon and Betty Moore Foundation 3790 Office of Science of the US Department of Energy DE-AC02-05CH11231 Funding was provided by NSF (#OCE1829831, #ABI1758974), the U.S. Department of Energy (#DE-SC0020173 and #248445), and the Gordon and Betty Moore Foundation (#3790). The work conducted by the U.S. Department of Energy Joint Genome Institute is supported by the Office of Science of the US Department of Energy under contract no. DE-AC02-05CH11231. An award from the Ohio Supercomputer Center (OSC) to Matthew B Sullivan supported computing resources used here. The funders had no role in study design, data collection and analysis, decision to publish, or preparation of the manuscript.

==============================
Background

Viruses influence global patterns of microbial diversity and nutrient cycles. Though viral metagenomics (viromics), specifically targeting dsDNA viruses, has been critical for revealing viral roles across diverse ecosystems, its analyses differ in many ways from those used for microbes. To date, viromics benchmarking has covered read pre-processing, assembly, relative abundance, read mapping thresholds and diversity estimation, but other steps would benefit from benchmarking and standardization. Here we use in silico-generated datasets and an extensive literature survey to evaluate and highlight how dataset composition (i.e., viromes vs bulk metagenomes) and assembly fragmentation impact (i) viral contig identification tool, (ii) virus taxonomic classification, and (iii) identification and curation of auxiliary metabolic genes (AMGs).

Results

The in silico benchmarking of five commonly used virus identification tools show that gene-content-based tools consistently performed well for long (≥3 kbp) contigs, while k-mer- and blast-based tools were uniquely able to detect viruses from short (≤3 kbp) contigs. Notably, however, the performance increase of k-mer- and blast-based tools for short contigs was obtained at the cost of increased false positives (sometimes up to ∼5% for virome and ∼75% bulk samples), particularly when eukaryotic or mobile genetic element sequences were included in the test datasets. For viral classification, variously sized genome fragments were assessed using gene-sharing network analytics to quantify drop-offs in taxonomic assignments, which revealed correct assignations ranging from ∼95% (whole genomes) down to ∼80% (3 kbp sized genome fragments). A similar trend was also observed for other viral classification tools such as VPF-class, ViPTree and VIRIDIC, suggesting that caution is warranted when classifying short genome fragments and not full genomes. Finally, we highlight how fragmented assemblies can lead to erroneous identification of AMGs and outline a best-practices workflow to curate candidate AMGs in viral genomes assembled from metagenomes.

Conclusion

Together, these benchmarking experiments and annotation guidelines should aid researchers seeking to best detect, classify, and characterize the myriad viruses ‘hidden’ in diverse sequence datasets.

Introduction

Viruses that infect microbes play significant roles across diverse ecosystems. For example, in marine systems, viruses are now broadly recognized to modulate biogeochemical cycles via lysis (e.g., heterotrophic prokaryotes lysis) (Fuhrman, 1999; Wilhelm & Suttle, 1999), alter evolutionary trajectory of core metabolisms via horizontal gene transfer (Sullivan et al. 2006), and impact the downward flux of carbon that helps the oceans buffer us (humans) against climate change (Guidi et al., 2016; Lara et al., 2017; Laber et al., 2018; Kaneko et al., 2019).

Viromics (viral metagenomics) has helped further our understanding of marine viral genomic diversity, and ecosystem roles (Mizuno et al., 2013; Anantharaman et al., 2014; Coutinho et al., 2017; Nishimura et al., 2017a; Ahlgren et al., 2019; Haro-Moreno, Rodriguez-Valera & López-Pérez, 2019; Ignacio-espinoza, Ahlgren & Fuhrman, 2019; Luo et al., 2020). Ecologically, we now have global ocean catalogs approaching 200K dsDNA viruses that have been used to provide ecological maps of community structure and drivers (Mizuno et al., 2013; Brum et al., 2015; Roux et al., 2016; Coutinho et al., 2017; Gregory et al., 2019), and to formally (Gregory et al., 2019) and empirically (Gregory et al., 2019; Haro-Moreno, Rodriguez-Valera & López-Pérez, 2019) demonstrate that these viral populations represent species. Biogeochemically, viral roles in biogeochemistry now appear more nuanced as viruses impact biogeochemical cycling not only by lysing their microbial hosts as has been studied for decades (Fuhrman, 1999; Wilhelm & Suttle, 1999), but also by reprogramming cellular biogeochemical outputs either broadly through viral take-over and infection (the ‘virocell’) or more pointedly by expressing ‘auxiliary metabolic genes’ (AMGs) during infection that alter specific metabolisms of the cell (Breitbart et al., 2007; Lindell et al., 2007; Rosenwasser et al., 2016; Howard-Varona et al., 2020). While AMGs were initially discovered in cultures [e.g., photosynthesis genes (Mann et al., 2003)], viromics has drastically expanded upon these to now also include dozens of AMGs for functions across central carbon metabolism, sugar metabolism, lipid–fatty acid metabolism, signaling, motility, anti-oxidation, photosystem I, energy metabolism, iron–sulfur, sulfur, DNA replication initiation, DNA repair, and nitrogen cycling (Clokie et al., 2006; Sharon et al., 2007; Dinsdale et al., 2008; Millard et al., 2009; Wommack et al., 2015; Hurwitz, Brum & Sullivan, 2015; Roux et al., 2016; Breitbart et al., 2018; Roitman et al., 2018; Ahlgren et al., 2019; Gazitúa et al., 2020; Kieft et al., 2020; Mara et al., 2020).

Beyond the oceans, viromics is also providing novel biological insights in e.g., humans (Lim et al., 2015; Norman et al., 2015; Reyes et al., 2015; Aiemjoy et al., 2019; Clooney et al., 2019; Fernandes et al., 2019; Gregory et al., 2020b), soils (Zablocki, Adriaenssens & Cowan, 2015; Trubl et al., 2018; Jin et al., 2019; Li et al., 2019; Santos-Medellin et al., 2020), and extreme environments (Adriaenssens et al., 2015; Scola et al., 2017; Bäckström et al., 2019; Zhong et al., 2020). Together these studies provide a baseline ecological understanding of viral diversity and functions across diverse ecosystems.

Critically, however, viromics remains an emerging science frontier with methods and standards very much in flux. To date, standardization efforts have included (i) establishing quantitative data generation methods (Yilmaz, Allgaier & Hugenholtz, 2010; Duhaime et al., 2012; Hurwitz et al., 2013; Solonenko & Sullivan, 2013; Conceição-Neto et al., 2015; Roux et al., 2017), and (ii) analytical benchmarks for read pre-processing, metagenomics assembly, and thresholds for relative abundance, read mapping and diversity estimation (Brum et al., 2015; Gregory et al., 2016; Roux et al., 2017). Further, though not from viral particle derived metagenomes (viromes), related efforts have also been made to provide recommendations for how best to analyze viruses in bulk metagenomic samples (Paez-Espino et al., 2016; Paez-Espino et al., 2017; Dutilh et al., 2017; Emerson et al., 2018).

Here we contribute to this growing set of community-driven benchmarks and guidelines. Specifically, we use in silico datasets that mimic viromes (specifically of dsDNA viruses) and bulk metagenomes with varied amounts of non-virus ‘distractor’ sequences to evaluate (i) options for viral identification, (ii) genomic fragment sizes for viral classification via gene-sharing networks, as well as (iii) provide guidelines for best practices for the evaluation of candidate AMGs.

Material and Methods

Dataset

Datasets used in this study included genomes from: (i) NCBI virus RefSeq v.203 (released December 2020); to avoid including the same genomes used in any of the viral identification tools and vConTACT v2, we chose only complete genomes released after May 2020 (1,213 genomes, see Table S1), (ii) Bacteria RefSeq v.203 (174,973,817 genomic scaffolds), (iii) archaea RefSeq v.203 (2,116,989 genomic scaffolds), (iv) NCBI plasmids v.203 (1,339,171 genomes), and (iv) Human GRCh38 as the eukaryotic dataset. All datasets were downloaded from NCBI RefSeq, last accessed in December 2020 (the links are listed in the ‘availability of data and materials’ section below). In addition, we also added ∼142 dsDNA cyanophage genomes to include a set of closely related genomes, as can sometimes be obtained from viromics experiments (Table S1) (Gregory et al., 2016).

Dataset simulation

in silico simulations were adapted to benchmark the viromics pipelines for (i) virus identification and (ii) virus classification. The overall framework of dataset simulation strategies is shown in Fig. 1. The simulation created four randomized subsampled datasets that were further fragmented to mimic fragmented assemblies of viromes and bulk metagenomes for viral contig identification and classification. An in-house script was used to split eukaryotic, prokaryotic, and plasmid sequences into non-overlapping fragments of different lengths, i.e., L = 500 bp, 1 kbp, 3 kbp, 5 kb, 10 kbp, and 20 kbp. Non-overlapping fragments from each sequence category (viral, prokaryotic, eukaryotic, plasmid) were then combined to reflect mock communities’ composition (see below). These mixed datasets were used to benchmark viral contigs identification tools (Fig. 1), while benchmarking of virus classification was performed only on fragmented sequences from viral RefSeq (Fig. 1).

Figure 1 The framework of dataset simulation strategies.

First, the viral RefSeq, prokaryote, eukaryote, and plasmid genome sequences were fragmented, from 5′ to 3′ end direction, into non-overlapping fragments of different lengths, i.e., L = 500 bp, 1 kbp, 3 kbp, 5 kbp, 10 kbp, and 20 kbp fragments. Then, these non-overlapping fragments were randomly sub-sampled to obtain simulated input datasets. For virus identification analysis, these simulated datasets were designed to resemble mock communities with different ratios of viral, prokaryote, eukaryote and plasmid sequences, i.e., virome_1 (10:1:0.1:0.01), virome_2 (10:1:0.01:1), bulk_1 (1:10:0.1:10), and bulk_2 (1:10:1:1). For viral classification analysis, simulated inputs were exclusively composed of fragmented viral genomes.

Mock communities

The four mock communities (with four replicates for each dataset) were randomly constructed to include different virus, prokaryotic, eukaryotic and plasmid sequences in ratios (Fig. 1) that varied to represent communities enriched in viral genomes (Roux et al., 2015), i.e., ‘virome_1 (up to 20,021 sequences; ratio, 10:1:0.1:0.001)’ and ‘virome_2 (up to 20,021 sequences; ratio, 10:1:0.01:1)’ or cellular genomes, i.e., ‘bulk_1 (up to 270,271 sequences; ratio, 1:10:0.01:10)’ and ‘bulk_2 (up to 22,035 sequences; ratio, 1:10:1:1)’ (Fig. 1). To further investigate the potential source of errors in viral contigs identification, we also fragmented datasets consisting only of archaea, plasmid and eukaryotes (human; Fig. 1).

Viral contig identification

The tools used for viral identification included VirSorter (Roux et al., 2015), MetaPhinder (Jurtz et al., 2016), MARVEL (Amgarten et al., 2018), DeepVirFinder (Ren et al., 2019), and VIBRANT (Kieft, Zhou & Anantharaman, 2020). Different cutoffs were applied for each of the tools, as follows, (i) we used two different versions of VirSorter, v1.0.5 and v1.10. VirSorter v1.05 used the viromedb database, while VirSorter v1.10 included the same viromedb database, as well as the Xfam database (Emerson et al., 2018; Gregory et al., 2019). For VirSorter (version 1.0.5 and 1.1.0; with ‘–db 2 –virome –diamond’), different cutoffs were used and compared: either all VirSorter predictions were considered as viruses (categories 1–6), or, only predictions of categories 1, 2, 4, and 5 was considered as viruses. For DeepVirFinder (version 1.0), we used three score cutoffs: ≥0.7, ≥0.9 and ≥0.95 and p-values ≤ 0.05. For MARVEL (version 0.2), two score cutoffs were used: ≥70% and ≥90%. Finally, for VIBRANT we used two different versions, i.e., version 1.1.0 and version 1.2.0; with ‘-virome’ and no ‘-virome’ setting, and MetaPhinder, default settings were used. The performance metrics to evaluate the efficiency of each tool were: (1) MCC=TP×TN−FP×FNTP+FPTP+FN×TN+FPTN+FN

Where MMC is Matthews’s correlation coefficient, TP is true positive, TN is true negative, FP is false positive, and FN is false negative. MCC values range between −1 to 1, with 1 indicating perfect efficiency (Chicco & Jurman, 2020). (2) Recall=TPTP+FP

Where TP is true positive, and FP is false positive. (3) False−discoveryrate=FPFP+TN

Where FP is false positive, and TN is true negative. (4) Accuracy=TP+TNTP+TN+FP+FN

Where TP is true positive, TN is true negative, FP is false positive, and FN is false negative. (5) F1=2TP2TP+FP+FN

Where TP is true positive, TN is true negative, FP is false positive, and FN is false negative. (6) PVV=TPTP+FP

Where PVV is positive predictive value, TP is true positive, TN is true negative, FP is false positive, and FN is false negative. (7) Specificity=TNTN+FP

Where TN is true negative, and FP is false positive.

Statistical analysis

A Wilcoxon test was used to compare the overall performance of viral identification, on the basis of fragment length (with 20 kbp as a reference group), including MCC, recall, false discovery, accuracy, F1, PVV, and specificity. The analysis was done using the R program (https://www.r-project.org/).

Viral classification

To evaluate the impact of fragmented assembly on a gene-sharing network-based viral classification, we leveraged vConTACT v2 (Jang et al., 2019) and used fragmented viral RefSeq genomes of different lengths (i.e., 500 bp, 1 kbp, 3 kbp, 5 kbp, 10 kbp, and 20 kbp) with default parameters. Furthermore, we also applied vConTACT v2 to complete genomes as a control dataset. It is worth noting that vConTACT v2 (originally) uses RefSeq v.85 (now has been updated to RefSeq v.99) as a reference database and manually validated ICTV taxonomies (ICTV Master Species List v1.3- February 2018) (Jang et al., 2019). The metrics used were those of Jang et al. (2019) including: (i) accuracy (Acc), (ii) clustering-wise separation (Sep), (iii) the positive predictive value (PPV), (iv) clustering-wise sensitivity (Sn), (v) cluster-wise separation (Sepcl), and (vi) complex (ICTV taxonomy)-wise separation (Sepco). The formulas are available in Jang et al. (2019).

In addition to vConTACT v2, we also evaluated the impact of fragmented assembly on viral classification using VPF-class (protein family based) (Pons et al., 2021), VipTree (genome-wide similarity-based) (Nishimura et al., 2017b), and VIRIDIC (BLASTN-based) (Moraru, Varsani & Kropinski, 2020). To evaluate the result, for VPF-class, we used taxonomic assignation of fragments with confidence score (CS) of ≥0.2 and membership ratio (MR) of ≥0.2, that have been reported to result in 100% of accuracy (Pons et al., 2021). For VIRIDIC and ViPTree, since no taxonomic assignation is automatically generated, we used the similarity and distance matrices provided by these tools to evaluate their performance on fragmented genomes, by comparing the similarity/distances obtained from genome fragments to the ones obtained from complete genomes (Nishimura et al., 2017b; Moraru, Varsani & Kropinski, 2020).

AMG curation analysis

Recommendations and best practices for AMG curation were based on a survey of the recent AMG literature, including especially Roux et al. (2016), Enault et al. (2017), Breitbart et al. (2018), Kieft et al., 2020. To illustrate the major challenges in the AMG identification process, we used DRAMv (Shaffer et al., 2020) to identify candidate AMGs in virus genomes from (Emerson et al., 2018; Mara et al., 2020). The following parameters were used: AMGs score of 1–3 and AMG flag of -M and -F. To verify the functional annotation of the candidate AMGs, we manually checked the genomic context of the viral contigs, i.e., the annotation of the neighboring genes (especially the presence of viral hallmark and viral-like genes), and the position of AMG with respect to the contig’s edge. Next, we then manually looked for the presence of promoter/terminator regions using BPROM (Linear discriminant function (LDF) >2.75; (Richardson & Watson, 2013), and ARNold (default setting; (Macke et al., 2001)). Conserved regions and active sites in the protein sequences were analyzed using PROSITE (Sigrist et al., 2013) and HHPred (Zimmermann et al., 2018) using the PROSITE collection of motifs (ftp://ftp.expasy.org/databases/prosite/prosite.dat), and PDB_mmCIF70_14_Oct (default) databases, respectively. For protein structural similarity, we used Phyre2 (confidence >90% and 70% coverage; (Kelly et al., 2015)), and predicted quaternary structures using SWISS-MODEL with a Global Model Quality Estimation (GMQE) score above 0.5 (Waterhouse et al., 2018). Eventually, we selected one representative example for different typical cases of either genuine AMGs or false-positive detections, which are visualized using genome maps drawn with EasyFig (Sullivan, Petty & Beatson, 2011).

Results and Discussion

Establishment of mock communities for in silico testing

We first benchmarked and compared strategies for identification of viruses across different types of metagenomes. Researchers have identified viruses from virus-enriched metagenomes (viromes), as well as bulk and/or cellular metagenomes that are typically dominated by prokaryotic or eukaryotic sequences, all with variable representation of other mobile elements (e.g., plasmids and transposons). We thus established mock community datasets that included viral, prokaryotic, eukaryotic and plasmid sequences in varied ratios to represent a diversity of datasets likely to be encountered in environmental samples (Fig. 1).

Briefly, two mock communities represented viromes and two represented bulk metagenomes, with ratios of virus: prokaryote: eukaryote: plasmid sequences as follows: ‘virome_1’ ratio = 10:1:0.1:0.001, ‘virome_2’ ratio = 10:1:0.01:1, ‘bulk_1’ ratio = 1:10:0.01:10 and ‘bulk_2’ ratio = 1:10:1:1 (see Methods and Materials for details, Fig. 1). Clearly benchmarking are needed for other viral types since our focus here was dsDNA viruses. It is also worth noting that to better mimic viral populations in natural system, we complemented RefSeq genomes by specifically adding closely related genomes to the datasets from the only such deeply sequenced ‘reference’ dataset available (cyanophages (Gregory et al., 2016), see Materials and Methods). To reflect the fragmented assembly typically obtained from short-read metagenomes, we extracted random subsets of varying length (500 bp–20 kbp) from these genomes, which were then combined at different ratios. Importantly, for viral RefSeq dataset, we only consider recent viral genomes submitted after May 2020, this to avoid including genomes that were used in training of any of the tools benchmarked here.

Comparison of viral identification tools

Several bioinformatic analysis tools have been developed to identify viruses from metagenomes (Table 1), using three major approaches: (i) similarity to known viruses, (ii) gene content/features, and (iii) k-mer frequency (i.e., nucleotide composition). Here, we first compared the performance of the most commonly used viral identification tools: VirSorter (Roux et al., 2015), MetaPhinder (Jurtz et al., 2016), MARVEL (Amgarten et al., 2018), DeepVirFinder (Ren et al., 2019), and VIBRANT (Kieft, Zhou & Anantharaman, 2020) against our suite of mock communities. We attempted to include two additional tools PHASTER (Arndt et al., 2017), and VirMiner (Zheng et al., 2019)—but these did not scale and were eventually not included in the test results presented here. A range of parameters and cutoffs (see Methods and Materials for details) were used to assess the performance of each tool across different fragment sizes (ranging 500 bp–20 kbp). Tool performance was evaluated using the following metrics: (i) ‘efficiency’, assessed using Matthews correlation coefficient, an overall statistic for assessing the recall and false-discovery, which this measure (MCC) offers a more informative and truthful evaluation than accuracy and F1 score (Chicco & Jurman, 2020), (ii) recall, (iii) false-discovery rate, (iv) accuracy, (v) F1, (vi) PVV, and (vi) specificity (see the formulas in Materials and Methods).

Table 1 The comparison of the commonly-used viral identification tools.

Tool	Approach	Basic mode	Ability to process modern-scale (viral) metagenomes scalability	Reference	
VirSorter	Gene-content-based tool. Features include enrichment in viral-like genes, depletion in PFAM hits, enrichment in short genes, and depletion in coding strand changes	Permissive cutoff category 1–6
Conservative category 1245 Setting for -virome, enable virome decontamination mode	Yes	Roux et al. (2015)	
MARVEL	Gene-content-based tool. Features include average gene length, average spacing between genes, density of genes, frequency of strand shifts between neighboring genes, ATG relative frequency, and fraction of genes with significant hits against the pVOGs database	Permissive cutoff ≥70% Conservative ≥90%	Yes	Amgarten et al. (2018)	
VIBRANT	Gene-content-based tool. Features include ratio of KEGG hits, ratio of VOG hits, ratio of PFAM hit, as well as presence of key viral-like genes (e.g., nucleases, integrase, etc.)	Default	Yes	Kieft et al. (2020)	
MetaPhinder	Integrated analysis of BLASTn hits to a, bacteriophage database, no gene prediction or amino acid-level comparison	Default	Yes	Jurtz et al. (2016)	
DeepVirFinder	K-mer based similarity to viral and host databases, no gene prediction or amino acid-level comparison	Permissive cutoff score ≥0.7, Medium ≥0.90, Conservative ≥0.95, and p-value ≤ 0.05	Yes	Ren et al. (2019)	
VirMiner	Gene-content-based tool. Features include ratio of hits to KO, ratio of hits to POGs, ratio of hits to PFAM, and presence of hallmark genes	Default. Web server: http://147.8.185.62/VirMiner/	No	Zheng et al. (2019)	
PHASTER	Gene-content-based tool. Features include number of phage-like genes, with additional annotation of e.g., tRNA to better predict prophage boundaries	Default. Web server: https://phaster.ca	No	Arndt et al. (2017)	

Overall, we found that viral contigs were better identified (increased efficiency, MCC) as fragment sizes increased, and this was true for all tools evaluated (Fig. 2 and Figs. S1–S4, Wilcoxon test, p-value ≤ 0.0001). However, tools based on gene content, i.e., VIBRANT, MARVEL, and VirSorter (v1.05 and v1.10) decreased sharply in efficiency (MCC) with input sequences ≤3 kbp and particularly ≤1 kbp (Figs. 2E–2H), whereas this decrease was less pronounced for DeepVirFinder (k-mer based) and MetaPhinder (BLASTN based) at these smaller size ranges (MCC values ∼0.20−0.625; Fig. 2E–2H). Notably, the trade-off of this efficiency was a higher false-discovery that reached as much as ∼5% for virome and ∼80% for bulk samples in our mock communities as compared to <1% when longer fragments were used (Fig. 2M–2P).

Figure 2 The viral identification analysis across datasets.

The viral identification analysis across datasets. (A–D) Pie-charts of the composition of the datasets depicted the different fragment sizes of the (i) virome_1, (ii) virome_2, (iii) bulk_1, and (iv) bulk_2. (E–H) The viral identification efficiency was calculated as Matthew’s correlation coefficient (MCC), where 1 represents perfect efficiency, (I–L) Percent of recall (%), and (M–P) Percent of false-discovery (%) of DeepVirFinder, MetaPhinder, MARVEL, VIBRANT, and VirSorter. For DeepVirFinder, three cutoffs were evaluated, i.e., score ≥0.7, ≥0.9, ≥0.95, and p-value ≤ 0.05. For MARVEL, two cutoffs were used, i.e., scores of ≥70% and ≥90%. Next, we use two different versions of VirSorter, i.e., v1.05 and v1.10, and two cutoffs, i.e., category 1, 2, 3, 4, 5, 6 and category 1, 2, 4, 5. The upper error bars represent the mean of the replicates.

We next explored how permissive versus conservative parameter cutoffs impacted viral identification based on permissive and conservative cutoffs recommended for each tool (Roux et al., 2015; Jurtz et al., 2016; Amgarten et al., 2018; Ren et al., 2019; Kieft, Zhou & Anantharaman, 2020) (see Materials and Methods, and Fig. 2). As expected, ‘conservative’ thresholds led to lower recall and lower false-discovery than ‘permissive’ for all tools (Fig. 2). This illustrates the trade-off that researchers are faced with maximizing viral identification (especially for fragment sizes ≤ 3 kbp) using ‘permissive’ cutoffs and/or tools not based on gene content will almost always be associated with a higher rate of false-discovery. Ultimately, the initial research question of the study has to be considered to make the decision of which type of cutoffs to use.

Finally, we evaluated whether false-positive detections were associated with specific types of non-viral sequences, including other mobile genetic elements and ‘novel’ microbial genomes. To this end, we generated datasets composed only of archaea, plasmid, or eukaryotic sequences, and measured false-discovery rates across the viral identification tools (Fig. S3). It is important to note however that, to our knowledge, there is currently no ‘clean’ plasmid database that is not also containing phages/viruses’ genome. Therefore, our benchmark is based on a cleaning based on ‘complete’ plasmid/phages, and primarily looking at how genome fragmentation impacts the delineation of plasmid vs phage. Most tools showed an especially high false-discovery rate for plasmid and/or eukaryotic sequences, including VIBRANT v.1.2.0 when using the virome flag (highest in eukaryote up to > 90% false-discovery, while other version of VIBRANT is less affected), MetaPhinder (highest in plasmid up to >40% false-discovery), MARVEL (up to ∼20% false discovery for plasmid dataset), and VirSorter when using the virome flag (up to ∼24% false-discovery in eukaryote datasets) (Fig. S3). This suggests the data used to train these tools may have under-represented eukaryotic and/or plasmid sequence and highlights the importance of including diverse non-viral sequences in a balanced training set when establishing machine-learning based viral contig detection tools, as previously highlighted (Ponsero & Hurwitz, 2019; Kieft, Zhou & Anantharaman, 2020). Overall, two tools stand out in terms of maintaining the lowest false-discovery across the datasets: gene-content based VirSorter (conservative cutoff) and MARVEL (score ≥90%).

Together these comparisons suggest that viral identification efficiency increases with fragment length, and almost all tools are able to identify true viral contigs of 10 kbp or longer. At length > 3 kbp, ‘gene-content based tools’ are able to maximize viral recall and minimize false discovery at either permissive or conservative cutoffs, with VirSorter and MARVEL performing best under conditions where ‘distractor genomes’ (e.g., eukaryote, DPANN-archaea or plasmids) are expected to be prevalent. For researchers specifically aiming to identify short (<3 kbp) viral genome fragments, k-mer based tools (DeepVirFinder) and BLAST-based tool (MetaPhinder) would be the preferred choices, although while being aware of the potential high rate of false-positive detections, especially in samples where distractor genomes are expected to be prevalent.

Virus classification using fragmented data in gene-sharing networks

Once contigs from metagenomic assemblies are identified as viral, the next challenge a researcher faces is to determine what kind of virus they represent. Gene-sharing network analytics have emerged as a means to semi-automate such classification, and taxonomic assignations for whole genomes are robust even when the network includes varying amounts of fragmented genomes (Jang et al., 2019), but no studies have evaluated the taxonomic assignations of fragmented genomes themselves. Because viral genomes assembled from metagenomes are often partial, we sought to better understand how gene-sharing network approaches would perform for metagenome-derived viral sequences at various fragment lengths.

To answer this question, we first established a dataset of known genomes and then fragmented it to five fragment sizes that are commonly obtained from virome assemblies (Roux et al., 2017). Next, we evaluated the accuracy of taxonomic assignments for the variously sized genome fragments against those from complete genomes. Our results showed the percentage of sequences accurately assignable to specific viral taxa increased with fragment length. Specifically, the percentage of sequences clustered in a vConTACT v2 gene-sharing network increased from 61% to >80% from 3 kb to fragment to complete genomes (Fig. 3A). This difficulty in robustly integrating short genome or genome fragments in a gene-sharing network is further illustrated by the network topology itself, which shows a much higher fragmentation of the network for 3 kb fragment compared to complete genomes, accompanied by an inflated number of ‘new VCs’ and a higher number of unclustered sequence (whether outlier, overlapping, or singleton, Fig. S5). In addition to this lower rate of clustered sequences, short fragments also displayed a reduced percentage of sequences assigned to the correct genus (Fig. S6) and overall lower performance across all vConTACT v2 metrics tested (Fig. S7). This is consistent with the original vConTACT v2 benchmark which also noted that accurate classification was challenging to achieve for short complete genomes, i.e., genomes ≤ 5 kb (Jang et al., 2019). Hence, short fragments (<10 kb) may not be informative enough in terms of gene content to be robustly placed in a gene-sharing network and may artificially form ‘new’ virus clusters.

Figure 3 Viral classification analysis.

(A) Percentage of the input sequences in a vConTACT v2 cluster and (B) Percentage of sequences assigned by VFP-class to a genus. The performance of VPF-class was calculated using confidence score (CS) and membership ratio (MR) thresholds of ≥0.2 (Pons et al., 2021).

Currently, beyond vContact2, most viral classification tools such as VIRIDIC and VipTree have also been optimized to classify full viral genomes (Nishimura et al., 2017b; Moraru, Varsani & Kropinski, 2020). We thus sought to evaluate whether this decrease in performance with short fragments was a specificity of gene-sharing networks or was also observed for other taxonomic classification approaches. To test this, we performed similar comparisons of taxonomic assignment for varying genome fragment lengths using other viral classification tools including VipTree (genome-wide similarities-based), VIRIDIC (BLASTN-based), and VPF-class (protein family based). The general results show that the performance of these tools also increased with fragment size (Fig. 3B, Fig. S6, Fig. S8). For VPF-class, the percentage of sequence with a taxonomic assignation increased from ∼46% for 3 kbp fragments to ∼82% for 20 kbp (Fig. 3B), while the percentage of sequences assigned to the correct genus also increased with sequence length (Fig. S6B). For ViPTree and VIRIDIC, an increase in performance was also observed from 3 kbp through 20 kbp (Fig. S8). Together these results suggest genome fragmentation negatively impact virus taxonomic classification for all common approaches, with only longer genome fragments (≥10 kbp) providing sufficient information for an accurate and meaningful taxonomy assignment.

Auxiliary metabolic gene or not, that is the question

As sequencing technology and assembly algorithms improve, the increasing genomic context of uncultivated viruses provides an invaluable window into our ability to identify novel virus-encoded auxiliary metabolic genes, or AMGs. Problematically, however, until complete virus genomes are available, robustly identifying metabolically interesting genes in assembled (viruses) sequences from metagenomes remains a challenge for the field (e.g., see re-analyses of past ‘AMG’ studies in Roux et al. (2013) and Enault et al. (2017)). There are two major challenges in AMG analysis. First, even the most highly purified virus particle metagenome includes some degree of cellular genomic fragments (Roux et al., 2013; Zolfo et al., 2019). Thus, it is critical to demonstrate that any candidate AMG is indeed virus-encoded and not derived from cellular ‘contamination’, which requires adequate genomic context. Second, standard sequence analysis cannot always determine whether a candidate AMG is involved in a metabolic pathway or instead associated with ‘primary’ viral functions such as genome replication or host lysis. Based upon previous work (Clokie et al., 2006; Sharon et al., 2007; Dinsdale et al., 2008; Millard et al., 2009; Wommack et al., 2015; Hurwitz, Brum & Sullivan, 2015; Roux et al., 2016; Breitbart et al., 2018; Roitman et al., 2018; Ahlgren et al., 2019; Gazitúa et al., 2020; Kieft et al., 2020; Mara et al., 2020), we propose guidelines to systematize the evaluation of candidate AMGs including: (i) virus identification and quality assessment, (ii) AMG identification, genomic context assessment, and functional annotation, and (iii) further investigation of putative AMGs function (Fig. 4).

Figure 4 Proposed workflow and curation step for AMG identification and validation.

The recommend ed steps of a candidate AMGs include, (i) viral contig identification and quality assessment, (ii) AMG identification, genomic context assessment, and functional annotation, and (iii) further investigation of putative AMGs function.

Virus identification and quality assessment

For AMG studies, we recommend using a combination of tools with strict quality thresholds to identify high-confidence virus sequences (Fig. 4, ‘Viral contigs identification’). For example, high-confidence sequences might be those identified by Virsorter (cat 1,2) and VirFinder/DeepVirFinder (score ≥ 0.9, p-value < 0.05). For length of the contig, while we have in the past used viral contigs ≥1.5 kbp for AMG detection (Hurwitz, Hallam & Sullivan, 2013; Hurwitz, Brum & Sullivan, 2015; Roux et al., 2016), improved sequencing and assembly capabilities offer the opportunity to be less permissive since smaller contigs increase the risk of false positives. Currently, we recommend increasing the minimum size threshold for AMGs work to ≥10 kbp, or those that are circular (and thus interpreted to be complete genomes). Complementary to virus identification tools, we recommend using ViromeQC (Zolfo et al., 2019) to evaluate virome contamination at the dataset level, and CheckV (Nayfach et al., 2020) to identify and remove host contamination based on gene content for individual sequences. Finally, for cases where integrated prophages are likely assembled in a contig including both a host and a viral region, we recommend using prophage-specialized tools such as PHASTER (Arndt et al., 2017) for more refined prophage/provirus identification and boundary demarcation.

You are confident you have a virus sequence, but does it include any candidate AMGs?

Next, candidate AMGs must be identified within the selected high-confidence viral contigs (Fig. 4 “Identification of candidate AMGs”). The key step in this process is to correctly interpret results from a functional annotation pipeline to distinguish genes involved in host metabolism from genes involved in the viral replication cycle, often based on existing ontologies or manually defined keywords (Breitbart et al., 2018). To further refine this candidate AMG identification, it has been proposed that metabolic genes associated with a KEGG metabolic pathway would constitute “Class I” AMGs (i.e., highest confidence) while metabolic genes not directly included in a metabolic pathway (e.g., transport function) would represent “Class II” AMGs (lower confidence; (Hurwitz, Brum & Sullivan, 2015)). Importantly, depending on the definition one uses for ‘host metabolism’ vs ‘core viral functions’, some genes currently described in the literature as AMGs might not be legitimate AMGs, including some nucleotide-related genes (Kieft et al., 2020) or glycosyl transferases and glycoside hydrolases that are often used for surface attachment and entry (Shaffer et al., 2020). We thus recommend researchers to use the utmost caution when analyzing genes for which a true role and function remains uncertain and avoid systematically calling these simply “AMGs” without further qualifiers or caveats.

While prior AMG identification has often been done using manual inspection of the contigs functional annotation, there is opportunity now to advance towards a more systematic and semi-scalable approach to identify AMGs, with two new automated tools recently released. DRAM (Distilled and Refined Annotation of Metabolism), which is optimized for microbial annotation, but includes a ‘DRAM-v’ module for viruses, leverages expert-curated AMG databases for functional annotation and a two-component scoring system to assess the likelihood of a gene being encoded on a virus genome (Shaffer et al., 2020). Meanwhile VIBRANT, which is built for virus identification but also performs functional annotation, automatically curate KEGG-based annotations to highlight candidate AMGs associated to KEGG ‘metabolic pathways’ and ‘sulfur relay system’ categories (Kieft et al., 2020). Both tools thus provide a quick and automated way to obtain a list of candidate AMGs which nevertheless must be further analyzed to (i) verify that the candidate AMG is indeed encoded by a virus, and (ii) verify that the candidate AMG is indeed involved in a cellular metabolic pathway.

How do you recognize a candidate AMG that may not actually be virus-encoded?

Although automated annotation tools such as DRAM-v and VIBRANT are helpful in speeding up the identification of candidate AMG, any detailed ecological or evolutionary analysis of an AMG requires additional manual inspection of both genomic context and predicted functions. Here, we illustrate examples of typical “mistakes” made by automated tools (Fig. 4 ‘Genomic context assessment of candidate AMGs’).

First, two examples of sequences likely to be genuinely viral, either closely related to a known phage (contig_1, ‘likely viral’) or not (contig_2, ‘possible viral’) are presented in Fig. 4. These sequences are mostly composed of viral or unknown genes, with little to no ‘cellular-like’ gene outside of the single candidate AMG. Next to these however, are four examples of AMGs predicted yet unlikely to be viral (‘unlikely viral’ candidates). Contig_3 represents a sequence ∼120 kbp with dense, short genes, and no viral/viral-like genes. This sequence is likely to be a cellular genomic region, possibly a mobile genetic element, that could easily be mistaken for a phage by automated tools. Next in contig_4, the candidate AMG is surrounded by genes that reveal little evidence of belonging to a viral genome, but where VirSorter (categories 1 and 2) and/or VirFinder (score ≥ 0.9 and p-value < 0.05) suggest the contig overall is of viral origin. Conservatively, these genes AMGs should not be considered further due to the low contextual evidence of the region immediately surrounding the candidate AMG to be of viral origin. Finally, in contig_5, the candidate AMG is at the edge of the viral contig along with a tRNA and a phage integrase. This example likely represents the miscall of a prophage boundary, and the AMG-containing region is likely a small fraction of the host genome, where metabolic genes are much more common (Table S2). Overall, further examining the specific genomic context around each candidate AMG is highly recommended in order to identify false-positive detections, i.e., non-viral sequences wrongly considered as viral by automated tools. This is especially critical in AMG analysis because these non-viral regions, while overall rare among the entire set of sequences predicted as viral, will typically have a much higher probability of including genes annotated as metabolic, i.e., candidate ‘AMGs’. Hence, even a small number of contaminating sequences can substantially impact downstream AMG analyses.

How to recognize a true metabolic AMG?

As for their viral origin (see above), the predicted function of candidate AMGs will typically need to be refined beyond the results of automated functional annotation pipelines. While the ideal proof of function is through biochemical assay of the AMGs to support the in silico inferred function, this is laborious and time-consuming lab work, such that only a handful of AMGs known to date has been experimentally validated—psbA (Lindell et al., 2005; Clokie et al., 2006), pebS (Dammeyer et al., 2008), and glycoside hydrolase (Emerson et al., 2018). To provide scalable in silico evaluation of putative AMGs and guide future experimental validation, we recommend the following analyses (Fig. 4 ‘AMG functional analysis’).

First, deeper functional analyses should be conducted to assess, where possible, whether the AMG contains known conserved residues and active sites, as well as whether structural predictions are consistent with the sequence-based functional prediction (Fig. 4). The analysis of protein conserved regions and active sites can be done manually via inspection of sequence alignments, as well as semi-automatedly where possible using, e.g., PROSITE (Sigrist et al., 2013) and HHPred (Zimmermann et al., 2018). For protein structural predictions there are several available tools including Phyre2 (Kelly et al., 2015), SWISS-MODEL (Waterhouse et al., 2018), and I-TASSER (Yang & Zhang, 2015). Protein structure is known to be more conserved than primary protein sequence, thus enabling the annotation of more divergent proteins, as well as supporting other functional annotation pipelines (Kelly et al., 2015). Importantly, when interpreting results of predicted structures and structure-based similarity for candidate AMGs, one should verify that the predicted structure is consistent with the predicted biological function, but also consider the relationship between top hits, in which one would expect to have several of the top hits homologous to each other (Roux et al., 2016; Gazitúa et al., 2020). The latest recommended cutoffs for these functional annotation tools are provided in Table 2.

Table 2 Auxiliary metabolic genes (AMGs) curation guidelines.

Parameters	Analysis program	cutoffa	Note	Reference	
Viral assembled contig quality assessment	CheckV	Complete viral contigs	–	Nayfach et al. (2020)	
	ViromeQC	Default	–	Zolfo et al. (2019)	
AMG identification	VIBRANT	Default	–	Kieft et al. (2020)	
	DRAM-v	Default	Putative AMG criteria: AMG score 1–3, and -M and -F flag.	Shaffer et al. (2020)	
Conserved residues and active sites	PROSITE	Default	PROSITE collection of motifs (ftp://ftp.expasy.org/databases/prosite/prosite.dat) database	Sigrist et al. (2013)	
	HHPred	Default	database: PDB_mmCIF70_23_Jul	Zimmermann et al. (2018)	
	BPROM	Linear discriminant function (LDF) > 2.75	Bacteria σ-70 Promoters. In intergenic region or within 10 bp of start or stop of ORF	Richardson & Watson (2013)	
	TransTermHP	Confidence score > 90%	Terminators search	Kingsford, Ayanbule & Salzberg (2007)	
	ARNold	Default	Terminators search	(Macke et al., 2001)	
Protein structural	Phyre2	100% confident and ≥70% alignment coverage	Secondary and tertiary structure search	Kelly et al. (2015)	
	SWISS-MODEL	Global Model Quality Estimation (GMQE) score above 0.5	Quaternary structure	Waterhouse et al. (2018)	
	I-TASSER	Default	Protein structural	Yang & Zhang (2015)	
	TMHMM	Default	Transmembrane domain	Krogh et al. (2001)	
Synonymous and non-synonymous mutation	MetaPop	<0.3 represent strong purifying selection	Calculate the pN/pS	Schloissnig et al. (2013) and Gregory et al. (2020b))	
Notes.

a The recommendation cutoffs that can be used in each step of AMGs curation.

Evolutionary analyses can be used to assess whether selection appears to be acting on the viral gene homolog. For instance, the ratio of non-synonymous (p N) to synonymous polymorphisms (pS)—known as pN/pS—can be used to evaluate whether the candidate AMGs is under purifying selection as would be expected for a functional gene (Schloissnig et al., 2013; Roux et al., 2016). Pragmatically, pN/pS values can be calculated manually using tools designed specifically for analyzing micro- and macro-diversity in metagenomes (e.g., MetaPop; Gregory et al., 2020a).

Your AMG appears viral and predicted to be functional and involved in host cell metabolism, what is its ecological and evolutionary story to tell?

Until this point, the candidate AMGs have gone through a series of meticulous vetting steps resulting in putative AMGs that can be used for downstream analyses such as phylogeny, ecological analysis, and experimental functional assays. We provide recommendations for each as follows (Fig. 4 “Additional (optional) analysis of the putative AMGs”).

To assess the evolutionary history of AMGs, phylogenetic analysis is carried out on individual AMGs and their corresponding microbial homologs. Briefly, for each AMG, one first needs to obtain homologs via sequence similarity searches (e.g., BLAST vs an appropriate database), then do multiple sequence alignments (e.g., MAFFT (Katoh et al., 2002), assess for intragenic recombination (e.g., RDP4 software (Martin et al., 2015)), build phylogenetic trees (e.g., IQ-TREE (Nguyen et al., 2015), and visualize them (e.g., iTOL, (Letunic & Bork, 2019). With these data in-hand, each phylogenetic tree can be examined to determine the number of transfer events that have occurred between microbes and viruses, as well as the ‘origin’ of the AMGs within the cellular and viral sequences in the analyses (sensu (Sullivan et al., 2010)).

Bona fide AMGs also typically have an ecological story to tell. Currently, the abundance of AMGs is estimated by read mapping against the viral populations that contain those AMGs (Gazitúa et al., 2020). However, a more sophisticated approach, where possible, would be to use the evolutionary inferences and multiple sequence alignments to identify virus-specific ‘signatures’ in the sequences that could be read-mapped to differentiate viral from cellular contributions to the gene, transcript, or protein pool in any given natural community. While such analyses are quite rare, e.g., (Sharon et al., 2007; Tzahor et al., 2009) growing AMG datasets should empower researchers to address this question of the virus ‘AMG’ contributions. Further, as virus-host prediction capabilities improve (Edwards et al., 2016; Villarroel et al., 2016; Galiez et al., 2017; Emerson et al., 2018; Wang et al., 2020), there is opportunity to combine these with AMG predictions to build understanding of ecologically-critical nuances of virus-host interactions. Finally, viral AMGs are under very different selective pressures than their host homologues given their viro-centric roles during infection. Will functional validation reveal viral versions that are fundamentally different? On one side, we may expect viral AMGs to have subtle mutations that might impact their enzyme efficiency (e.g., mutations in the PEST domain of PsbA (Sharon et al., 2007)) or substrate preferences (Enav et al., 2018). On the other side, we may expect viruses to encode more efficient proteins with ‘new’ functions. An example here is cyanophage-encoded ‘PebA’, which was thought to be a divergent 15,16-dihydrobiliverdin: ferredoxin oxidoreductase (pebA), but experimentally was shown to combine the capabilities of two host enzymes, PebA and PebB, to directly convert biliverdin IXX α to phycoerythrobilin and was thus renamed to PebS, a phycoerythrobilin synthase (Dammeyer et al., 2008).

Together, we hope these guidelines provide best-practice standard operating procedures for scientists to identify and evaluate candidate AMGs, as well as an emerging roadmap for how best to robustly bring this more nuanced and under-studied component of virus-host interactions to light so that viruses can be better incorporated into ecosystem models.

Conclusions

While viromics has proven invaluable for revealing the roles of viruses across diverse ecosystems, the emergent field of viral ecogenomics is in a state of rapid flux, experimentally and analytically. Here, we add to recent best practices efforts by evaluating and providing benchmarking for identifying and classifying viruses from viral-particle-enriched and bulk metagenomes, as well as recommendations for best practices for studying virus-encoded auxiliary metabolic genes. These efforts addressed some critical issues in standard operating procedures for viral ecogenomics researchers. Similar efforts will be needed to establish best practices in studying new emerging types of analysis and data including micro-diversity of virus populations (Gregory et al., 2019), and long-read sequencing (Warwick-Dugdale et al., 2018; Zablocki et al., 2021). Further, technological and analytical opportunities are being developed to better capture ssDNA and RNA viruses, as well as to establish dsDNA viral activity (Moniruzzaman et al., 2017; Emerson et al., 2018; Roux et al., 2019; Sommers et al., 2019; Starr et al., 2019; Trubl et al., 2019; Callanan et al., 2020). Finally, though viral discovery is now performed tens to hundreds of thousands of viruses at a time, the ability to link these new viruses to their hosts is still limited. Improved in silico approaches, such as those based on BLAST similarity, k-mers (such as WIsH (Galiez et al., 2017), HostPhinder (Villarroel et al., 2016)), and VirHostMatcher (Wang et al., 2020)), and CRISPR-Cas (Paez-Espino et al., 2016) have been recently proposed to predict the potential hosts of uncultivated viruses, which still need to be thoroughly tested and benchmarked across a variety of dataset types and sizes. Moreover, predictions from these in silico prediction tools need to be complemented with robustly benchmarked, high-throughput experimental methods, e.g., epicPCR, viral tagging, Hi-C (Deng et al., 2014; Bickhart et al., 2019; Yaffe & Relman, 2020; Sakowski et al., 2021) to validate these predictions.

Supplemental Information

Supplemental Information 1 Additional metrics for viral identification analysis across datasets

(A) The viral identification accuracy, (B) F1, (C) PVV, and(D)specificity of DeepVirFinder, MetaPhinder, MARVEL, VIBRANT, and VirSorter on the different fragment sizes of the (i) virome_1, (ii) virome_2, (iii) bulk_1, and (iv) bulk_2 (the composition of the datasets depicted as pie-charts). For DeepVirFinder, three cutoffs were evaluated, i.e., score ≥0.7, ≥0.9, ≥0.95, and p-value ≤0.05. For MARVEL, two cutoffs were used, i.e., scores of ≥70% and ≥90%. Next, we use two different versions of VirSorter, i.e., v1.05 and v1.10, and two cutoffs, i.e., category 1, 2, 3, 4, 5, 6 and category 1, 2, 4, 5. The upper error bars represent the mean of the replicates (see more in Materials and Methods).

Click here for additional data file.

Supplemental Information 2 Results from viral identification analysis for VIBRANT with no virome flag

(A) MCC, (B) recall, (C) false discovery rate, (D) accuracy, (E) F1, (F) PVV, and (G) specificity of (i) virome_1, (ii) virome_2, (iii) bulk_1, and (iv) bulk_2 datasets. The upper error bars represent the mean of the replicates.

Click here for additional data file.

Supplemental Information 3 False-discovery rate of viral sequence identification tools when challenged with Archaea, Eukaryote and Plasmid datasets

In addition to the parameters used in (Fig. 2 and Fig. S1), two additional settings of VIBRANT, i.e., without virome flag, were added, and two VirSorter, i.e., without virome flag (standard setting) were also applied. A color gradient represents the viral identification tool. The upper error bars represent the mean of the replicates (see more in Materials and Methods).

Click here for additional data file.

Supplemental Information 4 Pairwise analyses of the combined viral identification tools efficiency parameters for varying genome sizes

Metrics used included (A) MCC, (B) recall, (C) false discovery rate, (D) accuracy, (E) F1, (F) PVV, and (G) specificity of virome_1, virome_2, bulk_1, and bulk_2 datasets. The Wilcoxon test was used to compare fragment sizes (non-parametric) of the overall performance of viral identification. The 20k dataset was used as a reference data (see more in Materials and Methods). Only significant comparison is shown as follows, *: p-value ≤0.05; **: p-value ≤ 0.01; ***: p-value ≤ 0.001; ****: p-value ≤0.0001.

Click here for additional data file.

Supplemental Information 5 Gene-sharing network topology and performances

(A) the gene-sharing network for genome 3 kbp size, and (B) complete-genome. (C) the count of new viral clusters and clusters with reference genomes, and (D) the percentage of the overlapping, outliers, and singletons.

Click here for additional data file.

Supplemental Information 6 Percentage of ‘correct’ genus-assignments

(A) vConTACT v2 and (B) VFP-class. The analysis was performed by evaluating the correct genus assignment in comparison to the complete genomes. Only the genus affiliated genomes were considered in the evaluation for vConTACT v2.

Click here for additional data file.

Supplemental Information 7 Comparison of the performances of viral gene-sharing network-based classification, vConTACT v2 across fragment size

(A) Accuracy (Acc), (B) Clustering-wise separation (Sep), (C) clustering-wise PPV (PPV), (D) clustering-wise sensitivity (Sn), (E) cluster-wise separation (Sep cl), and (F) complex-wise sensitivity (Sep co). Complete formula can be seen in Jang et al., (2019).

Click here for additional data file.

Supplemental Information 8 Performance evaluation of VipTree and VIRIDIC across fragment sizes

(A) ViPTree, and (B) VIRIDIC (Nishimura et al., 2017b; Moraru, Varsani & Kropinski, 2020). Because VIRIDIC and ViPTree do not provide a taxonomic affiliation per se but a distance matrix, the differences in performance between different fragment sizes were calculated by comparing the pairwise distances obtained from the fragment sizes with the distances obtained from the complete genomes for the same pair, expressed as percentage relative to the distance obtained from complete genomes. While these percentage reflects differences in the similarity matrix obtained with different fragment sizes for a given tool, these cannot be directly compared between tools or to the performance metrics used to assess the impact of genome fragmentation on the taxonomic classification obtained by vConTACT v2 and VPF-Class.

Click here for additional data file.

Supplemental Information 9 Virus genomes used in this study

Click here for additional data file.

Supplemental Information 10 AMG annotation table

Click here for additional data file.

We would like to thank Drs. Heather Maughan and Chistine Sun for comments on the structure of an early draft of the manuscript, Drs. Ho Bin Jang and Olivier Zablocki, as well as Mohamed M. Mohamed, and Funing Tian for the many constructive discussions.

Abbreviations

MCC Matthews’s correlation coefficient

Sn clustering-wise sensitivity

PPV the positive predictive value

Acc accuracy

Sepco complex (ICTV taxonomy)-wise separation

Sepcl cluster-wise separation

Sep clustering-wise separation

Additional Information and Declarations

Competing Interests

Author Contributions

DNA Deposition

Data Availability

Maria Consuelo Gazitúa is a founder and CEO of Viromica Consulting, Chile.

All the authors declare that they have no competing interests.

Akbar Adjie Pratama conceived and designed the experiments, performed the experiments, analyzed the data, prepared figures and/or tables, authored or reviewed drafts of the paper, and approved the final draft.

Benjamin Bolduc, Ahmed A. Zayed, Simon Roux and Matthew B Sullivan conceived and designed the experiments, analyzed the data, prepared figures and/or tables, authored or reviewed drafts of the paper, and approved the final draft.

Zhi-Ping Zhong, Jiarong Guo, Dean R. Vik and Maria Consuelo Gazitúa performed the experiments, analyzed the data, authored or reviewed drafts of the paper, and approved the final draft.

James M. Wainaina analyzed the data, authored or reviewed drafts of the paper, and approved the final draft.

The following information was supplied regarding the deposition of DNA sequences:

The raw data is available at Viral RefSeq v.203 for viral sequences, Bacteria RefSeq v.203, archaea RefSeq v.203, plasmids RefSeq v.203, human GRCh38 for eukaryote: GCA_000001405.15.

The 142 cyanophages’ genomes are available at GenBank: KJ019026–KJ019131, KJ019134–KJ019165, JN371768, and KF156338–KF156340.

Viral sequences: https://www.ncbi.nlm.nih.gov/genome/viruses/

Bacteria (https://ftp.ncbi.nlm.nih.gov/refseq/release/bacteria/), archaea (https://ftp.ncbi.nlm.nih.gov/refseq/release/archaea/), and plasmid (https://ftp.ncbi.nlm.nih.gov/refseq/release/plasmid/)

Human GRCh38 for eukaryote: https://www.ncbi.nlm.nih.gov/assembly/GCF_000001405.39/

The following information was supplied regarding data availability:

The scripts used for virus identification, classification, mock communities’ datasets, virus datasets, vConTACT2 input and AMG input files are available at Bitbucket: https://bitbucket.org/MAVERICLab/standard_viromics2/

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
