# Peer review of "Expanding standards in viromics: in silico evaluation of dsDNA viral genome identification, classification, and auxiliary metabolic gene curation"

_PeerJ, doi:10.7717/peerj.11447_

## Round 0.1 · original submission · Major Revisions

PeerJ will be happy to further consider your manuscript for publication once a number of concerns identified by the reviewers are addressed. Both reviewers found the manuscript to be well written, researching an important, timely topic. But they did identify a number of minor and major concerns that should be addressed in a revised version. In particular, the revision should better clarify the statistical tests used to support the analysis. There were also concerns regarding the databases and database versions used for the analysis. These concerns should be addressed in detail and if necessary, the analyses revised to ensure that they are appropriate to support this analysis. The use of the various analytical tools supporting these analyses also needs to be further clarified to ensure that their use is appropriate for the datasets under study. One suggestion to address several of these issues was to limit the scope of the paper to dsDNA bacteriophages. A continued emphasis on the virome in general, may require significant additional effort.

Reviewer 1 ·

Basic reporting

This is a well-written, clearly presented and cohesive manuscript, with appropriate references to literature, background and excellent figures. I found it an exciting read and an absolutely necessary piece of work to further the field of viromics.

For basic reporting I have only a couple of minor comments:
Line 120: There is no Table S1.
Line 123: typo “links is listed”
Line 129: Please deposit the in-house script in an appropriate repository (eg. GitHub) to make it available to other researchers.

Experimental design

Materials & Methods: Some of the figures (Fig3 and S1) are showing significance values, but no statistics are explained in the M&M section. Please add this information. From the figure legend of Fig3, it seems like there was an ANOVA performed with t-tests. Given that there are multiple values compared, what was the correction used? Were the data normally distributed? In the legend of Fig S1, the authors describe the use of a non-parametric comparison of means (Kruskal-Wallis with post-hoc Dunn test and Bonferroni correction). Why was this a different test? Please clarify.

Validity of the findings

I have a couple of major comments on the description and interpretation of the analyses and the databases used, that should be addressed to prevent misinterpretation of the usefulness of the benchmarked tools and suggested workflows.

Firstly, the virosphere comprises more than viruses of microbes (as mentioned in the first line of the introduction), and viruses of microbes are much more than just dsDNA viruses. But many of the tools and the analyses presented here, seem to be specific for dsDNA bacteriophages and to a lesser extent archaeal viruses (eg. VirSorter, vConTACT2). This is an important issue that needs to be clarified across the manuscript, as there are many publications and research groups exclusively working on, for example, RNA viromes or circular ssDNA viromes.
The mock dataset viral fractions are all derived from the viral RefSeq database which includes all types of viruses, ssRNA, dsRNA, ssDNA, dsDNA, segmented, linear, circular with vastly different genome sizes. How does this influence false discovery, recall, MCC? Or is the mock community solely derived from dsDNA viruses of microbes? There is a big difference between a 3 kb fragment of a 150 kb Caudovirales phage genome, a 3 kb geminivirus genome or a 5 kb microvirus bacteriophage genome. If the first one does not get recognised as viral, that is less of an issue than the latter two for which the fragment is the whole or >50% of the genome length.
It would be great if the authors could narrow down whether the benchmarking performed here can be generalised for all viromes and the whole virosphere, or whether it is for a specific subset, and if so which subset with which qualifiers.

My second main point follows on from the first point. The authors use a different version of the RefSeq database for classification using gene-sharing networks. This section of the manuscript should present more data backing up the claims of classification accuracy. Again, the RefSeq database contains all virus types the gene-sharing network from Jang et al 2019 only comprises bacterial and archaeal viruses. How can a virus be correctly classified if its reference genome is not represented in the network? What version of the taxonomy release was used and was it from the same year as the RefSeq release? At what ranks (species, genus, family, …, realm) were the sequences assigned and how did this rank affect the accuracy of the assignments? The authors show that the accuracy improved with increased fragment length, but what about short complete genomes? Maybe the short fragments were not robustly placed in the network because their reference counterparts weren’t present?

I find the AMG workflow and suggestions hugely helpful, but I come back to my previous points. I have only ever heard of AMGs in the context of bacteriophages. Are they relevant for other viruses? Is this workflow generalisable for all viromes?

Additional comments

General comment on the use of the viral RefSeq database: This database is a curated subset of all viral genomes available from GenBank and does not represent full viral diversity. NCBI maintains two validated datasets, RefSeq and neighbor genomes (https://www.ncbi.nlm.nih.gov/genome/viruses/about/assemblies/). The RefSeq database does not contain many viral genomes that are extremely similar to each other, and a mock community based on RefSeq alone might deviate from reality in which many viral populations may co-exist that are extremely similar. Since the benchmarking starts from assembled genome fragments, this is not necessarily an issue for the data analysis here, but the authors might want to address this as a limitation.

Minor semantics issue: the term viral sequence is used a lot, which can be anything (sequence reads, contigs, genomes). What the authors are really presenting here are assembled sequences and mock contig datasets, which should be made more clear to the audience.

Reviewer 2 ·

Basic reporting

In this manuscript the authors perform benchmarking analysis of different tools for virome analysis. Specifically, they test the ability of these tools to identify viral genomes in metagenomes/metaviromes, the ability to correctly assign taxonomic classification to these genomes, and to identify their auxiliary metabolic genes. The topic is extremely relevant as, to my knowledge, no studies have previously performed analysis such as the one presented by the authors. The experiments are well designed and the tutorial like structure of the discussion will likely be helpful to researchers trying to decide on the best tools for their specific needs. Nevertheless, some issues need to be addressed to make this manuscript suitable for publication.

Experimental design

Ln 110-114: The the authors did not analyze the performance of in silico host predictions methods for viruses, for which many approaches are currently available [1–4]⁠. I understand that this might not be within your scope and therefore I will not insist that you include such analysis (although I do believe it would increase the relevance of the manuscript). Nevertheless, given the importance of these methods for studies of viral ecology I believe at least a brief mention to these methods is necessary in the introduction section as well as an explanation for not including them.

Ln 118-123: The dataset selection is inadequate. The viruses from RefSeq are the same that were used to train the majority of the tools for viral identification. Meaning that tools such as VIBRANT, MARVEL and DeepVirFinder which rely on machine learning models should be able to easily identify these sequences as viral. To make it really challenging for these tools it is necessary to use a test set consisting of only sequences that are divergent from those in the training set. Ideally, to mimic real world scenarios, the use of uncultured viral sequences from metagenomes should be applied, preferably covering multiple different ecosystems, such as marine, freshwater, soil, gut etc. This would ensure that the results are representative of multiple ecosystems since the diversity of viruses within each of them should be different. To avoid any circular logic, I recommend that the authors perform manual identification of viruses from metagenomes from this ecosystems and then use these bona fide viral sequences, fragmented or not to estimate the performance of the tools.

Ln 174-183: There are other methods for viral genome classification that were not analysed [5–8]⁠⁠. Specifically those that rely on phylogenomic approaches. These are important and popular tools that are very likely to be affected by genome fragmentation and therefore a proper benchmarking of the performance of this tools should also be included.

Ln 174-183: Regarding the analysis with vCONTACT 2.0: The setup seems to follow a somewhat circular logic. If the “unknown” fragmented genomes are just a subset of the full RefSeq it should be relatively easy for this tool to classify the genome fragments because the whole genome version is also included in the set. Again, the use of metagenomic viruses would be a more realistic scenario. Furthermore, this type of analysis should be performed with leave-one-out cross validation to answer questions like: How is a genome fragment classified when there are no representatives of its taxa in the reference database?

Validity of the findings

Some of the findings are not valid due to the issues mentioned in the previous section. I will re-evaluate the validity of the findings once these issues are corrected.

Additional comments

Ln 72: Silveira et al., 2017 and Coutinho et al., 2017 actually refer to the same publication. And the correct citation is the one for Coutinho et al., 2017

Ln 106: Edwards et al., 2016 performed benchmarking for in silico host prediction approaches and not for diversity estimation

Ln 160: It would be useful to evaluate the performance of VIBRANT also without the –virome flag

Figure 2: You need to explicitly mention what the error bar represent in the bar plots. Also, from reading the methods I understood that there were only 4 mock communities and no replicates. Hence, it is unclear to me why there are error bars in this plot.

1. Galiez C, Siebert M, Enault F, Vincent J, Söding J. WIsH: who is the host? Predicting prokaryotic hosts from metagenomic phage contigs. Birol I, editor. Bioinformatics [Internet]. 2017;33:3113–4. Available from: https://academic.oup.com/bioinformatics/article-lookup/doi/10.1093/bioinformatics/btx383
2. Wang W, Ren J, Tang K, Dart E, Ignacio-Espinoza JC, Fuhrman JA, et al. A network-based integrated framework for predicting virus–prokaryote interactions. NAR Genomics Bioinforma [Internet]. Oxford University Press; 2020;2:505768. Available from: https://academic.oup.com/nargab/article/doi/10.1093/nargab/lqaa044/5861484
3. Edwards RA, McNair K, Faust K, Raes J, Dutilh BE. Computational approaches to predict bacteriophage–host relationships. Smith M, editor. FEMS Microbiol Rev [Internet]. 2016;40:258–72. Available from: https://academic.oup.com/femsre/article-lookup/doi/10.1093/femsre/fuv048
4. Villarroel J, Kleinheinz KA, Jurtz VI, Zschach H, Lund O, Nielsen M, et al. HostPhinder: A phage host prediction tool. Viruses. 2016;8:1–22.
5. Coutinho FH, Edwards RA, Rodríguez-Valera F. Charting the diversity of uncultured viruses of Archaea and Bacteria. BMC Biol [Internet]. BMC Biology; 2019;17:109. Available from: https://www.biorxiv.org/content/10.1101/480491v1.full
6. Meier-Kolthoff JP, Göker M. VICTOR: genome-based phylogeny and classification of prokaryotic viruses. Kelso J, editor. Bioinformatics [Internet]. 2017;33:3396–404. Available from: http://academic.oup.com/bioinformatics/article/doi/10.1093/bioinformatics/btx440/3933260/VICTOR-genomebased-phylogeny-and-classification-of
7. Rohwer F, Edwards R. The Phage Proteomic Tree: a Genome-Based Taxonomy for Phage. J Bacteriol [Internet]. 2002;184:4529–35. Available from: http://jb.asm.org/cgi/doi/10.1128/JB.184.16.4529-4535.2002
8. Nishimura Y, Yoshida T, Kuronishi M, Uehara H, Ogata H, Goto S. ViPTree: the viral proteomic tree server. Valencia A, editor. Bioinformatics [Internet]. 2017;33:2379–80. Available from: https://academic.oup.com/bioinformatics/article-lookup/doi/10.1093/bioinformatics/btx157

---

## Round 0.2 · Minor Revisions

I will be happy to accept your manuscript for publication after a few minor corrections are made. The reviewer comments detail all of the suggested edits that should be considered. As indicated in these comments, please pay particular attention to problems with references and figure legends. Also, please edit the text added in this current version to ensure that it is grammatically correct.

Reviewer 1 ·

Basic reporting

The writing is clear and professional. However, some of the new paragraphs have grammar mistakes that should be corrected prior to publication. This could be addressed by the editorial team.

Something went wrong with the referencing. For example: lines 273-277 all the references for the tools are wrong and jumbled, VirSorter referring to EasyFig, PHASTER to VIBRANT, etc. I think this is occurring all through the manuscript, but I haven’t cross-checked all entries. Please correct.

Experimental design

In my previous round of review, I asked about versioning of the RefSeq and the Taxonomy database, and whether they matched. The authors provided updated information with the date of use.
RefSeq Database & Table S1: The choice of input sequences is still confusing. The authors used 1,213 genomes of viruses of bacteria from the RefSeq database, but the number of classified dsDNA bacteriophage genomes published in May 2020 was 1,991 using a quick search of the ICTV Viral Metadata Resource (version May 1st, 2020). What happened to the >700 genomes? Were these deliberately omitted? Or did the authors use a previous version of the taxonomy release.


Lines 363-376: VIRIDIC is not a virus classification tool in itself, even though it is often used in virus classification. It is a pairwise whole genome comparison tool meant to be used with complete genomes on a user-selected subset of genomes suspected to be related, with user-specified thresholds for phage species (default 95%) and genus (default 70%) clustering. So using fragmented genomes to assess the performance of this tool in inherently flawed. If you use a reference genome and a fragment of that reference genome, the tool will state that they are not related enough to be classified together. There will be a similar issue with VipTree, which is also made and optimised to classify full genomes. I’m not familiar enough with VPF-class to know whether the same goes here.

Validity of the findings

The findings are novel and an important contribution to the literature.

The classification section was reworked and now focuses on the different classification of known genome fragments versus full genomes. This is a big improvement, and provides important information. However, this needs to be made clear in the abstract as well, where it seems like classification of any virome is discussed.

Additional comments

This is an important and timely contribution to the literature and I congratulate the authors on this excellent piece of work.

Reviewer 2 ·

Basic reporting

All my comments have been properly addressed. This manuscript is now suitable for publication.

Experimental design

The updated experimental design is adequate.

Validity of the findings

Conclusions are supported by the findings.

Additional comments

Ln 122: Change “retain” to “retained”

Ln 128: Change “been” to “be”

Ln 273-277: References in this section are switched

Caption for Figure S2 is wrong

Figure S4: This analysis refers to the combine results of all tools?

---

## Round 0.3 · accepted · Accept

Thank you for rapidly attending to the reviewer comments. I am now happy to accept your manuscript for publication.